# Severe COVID-19 in Non-Smokers: Predictive Factors and Outcomes

**DOI:** 10.3390/healthcare13091041

**Published:** 2025-05-01

**Authors:** Marko Djuric, Irina Nenadic, Nina Radisavljevic, Dusan Todorovic, Nemanja Dimic, Marina Bobos, Suzana Bojic, Predrag Savic, Tamara Nikolic Turnic, Predrag Stevanovic, Vladimir Djukic

**Affiliations:** 1Clinic for Anesthesiology and Intensive Care, University Clinical Hospital Center “Dr Dragisa Misovic-Dedinje”, No. 1, Str. Heroja Milana Tepica, 11030 Belgrade, Serbia; nenadicirina@gmail.com (I.N.); nemanjadimic@live.com (N.D.); marinamorisan@gmail.com (M.B.); subojic@yahoo.com (S.B.); predrag.stevanovic@med.bg.ac.rs (P.S.); 2Department of Anesthesiology, Reanimatology and Intensive Care, Faculty of Medicine, University of Belgrade, 11000 Belgrade, Serbia; 3Institute of Medical Physiology ”Richard Burian”, Faculty of Medicine, University of Belgrade, 11000 Belgrade, Serbia; nina_radisavljevic@outlook.com (N.R.); t.dusan@hotmail.com (D.T.); 4Clinic for Surgery, University Clinical Hospital Center “Dr Dragisa Misovic-Dedinje”, 11030 Belgrade, Serbia; predrag.savic@dragisamisovic.bg.ac.rs (P.S.); office@dragisamisovic.bg.ac.rs (V.D.); 5Department of Surgery, Faculty of Medicine, University of Belgrade, 11000 Belgrade, Serbia; 6Department of Pharmacy, Faculty of Medical Sciences, University of Kragujevac, 34000 Kragujevac, Serbia; tnikolict@gmail.com; 7N.A. Semashko Public Health and Healthcare Department, F.F. Erismann Institute of Public Health, I.M. Schenov First Moscow State Medical University, 119991 Moscow, Russia

**Keywords:** COVID-19, non-smoker, mortality, predictors, scores

## Abstract

**Background:** The COVID-19 pandemic revealed an unexpected pattern known as the “smoker’s paradox”, with lower rates of severe disease among smokers compared to non-smokers, highlighting the need for the specific investigation of disease progression in non-smoking populations. Objective: To identify early mortality predictors in non-smoking patients with severe COVID-19 through the evaluation of clinical, laboratory, and oxygenation parameters. **Methods:** This retrospective observational cohort study included 59 non-smokers hospitalized with COVID-19 between November and December 2020. Clinical parameters, laboratory findings, and respiratory support requirements were analyzed on Days 1 and 7 of hospitalization. ROC curves were constructed to assess the predictive value of the parameters. **Results:** The overall mortality rate was 54.2%. The seventh-day SOFA score showed the strongest predictive value (AUC = 0.902, *p* = 0.004), followed by pCO_2_ (AUC = 0.853, *p* = 0.012). Significant differences between survivors and non-survivors were observed in acid–base parameters, oxygenation indices, and hematological markers. Mortality rates varied significantly with ventilation type: 84.6% for IMV and 50% for NIMV, with no deaths in HFNC patients. **Conclusions:** Multiple parameters measured on Day 7 of hospitalization demonstrate significant predictive value for mortality in non-smoking COVID-19 patients, with the SOFA score being the strongest predictor. The type of respiratory support significantly influences outcomes, suggesting the importance of careful ventilation strategy selection.

## 1. Introduction

The 21st century has been marked by numerous severe infectious disease epidemics, including the 2003 outbreak of severe acute respiratory syndrome (SARS), the 2009 H1N1 influenza pandemic, the 2012 Middle East respiratory syndrome (MERS) outbreak, the 2013–2016 Ebola virus epidemic, and the 2015 Zika virus epidemic. Among these, the COVID-19 pandemic—caused by the novel SARS-CoV-2 virus—has emerged as one of the most significant global public health threats of modern times. The virus was first identified in December 2019 in Wuhan, People’s Republic of China, following the identification of clusters of atypical pneumonia cases among hospitalized patients. Since then, the COVID-19 pandemic has spread globally, resulting in substantial mortality. According to World Health Organization, more than 7 million confirmed deaths have been reported worldwide from January 2020 through February 2025, posing unprecedented challenges to healthcare systems [1,2,3].

The clinical presentation of COVID-19 is characterized by marked heterogeneity. While the majority of patients (81%) develop mild to moderate disease, a significant proportion progress to severe illness, with potentially life-threatening complications [4]. A meta-analysis encompassing 10,815 cases demonstrated that acute respiratory distress syndrome (ARDS), one of the most serious complications, occurs in 6.2% to 32.2% of patients, with a high mortality rate of 39% (95% CI: 23–56%) [5,6]. These findings underscore the importance of early identification of patients at high risk for severe disease progression and the implementation of appropriate therapeutic strategies.

Established risk factors for severe COVID-19 include advanced age, male sex, and the presence of comorbidities, particularly cardiovascular diseases, diabetes mellitus, obesity, chronic respiratory diseases, hepatorenal dysfunction, and malignancies [7,8]. However, contrary to expectations that tobacco use would contribute to severe clinical forms of COVID-19, epidemiological data revealed an unexpected pattern—lower hospitalization rates among active smokers [9,10]. This phenomenon, known as the “smoker’s paradox”, relates to the potential modulatory effects of nicotine on the angiotensin-converting enzyme 2 receptor (ACE2) and inflammatory response [11].

In light of current understanding regarding the potential impact of smoking status on COVID-19 progression, our research focuses on non-smoking patients with severe clinical presentation. This study aims to identify early mortality predictors in this specific population through the evaluation of clinical, hematological, biochemical, and inflammatory parameters, as well as disease severity scores. Identifying reliable predictors would enable more accurate risk assessment for fatal outcomes and timely optimization of therapeutic approaches.

## 2. Materials and Methods

### 2.1. Study Design and Participants

This retrospective observational cohort study was conducted during the first wave of the COVID-19 pandemic (between 12 November and 8 December 2020) and examined 59 non-smoking patients who were hospitalized for COVID-19 at the Clinical Hospital Center “KBC Dragiša Mišović-Dedinje” in Belgrade. The study was conducted according to the principles of the Declaration of Helsinki and was approved by institutional decision (no active or additional interventions were performed). The data for research purpose were extracted from medical database records; however, confidentiality and basic patients’ rights were respected, according to good clinical practices.

A SARS-CoV-2 infection was confirmed by a coronavirus (2019-nCoV) nucleic acid diagnostic kit (RT-PCR fluorescence probing), according to the manufacturer’s protocol (Sansure Biotech, Changsha, Hunan Province, China). Nasopharyngeal swab samples were collected and placed into a collection tube containing a preservation solution protecting the virus. The study specifically included non-smokers aged 18 years and older with confirmed SARS-CoV-2 infections. Patients were excluded if they were current or former smokers or if they had incomplete medical records.

Based on COVID-19 disease severity at hospital admission, patients were classified into one of four groups. Patients with very mild disease displayed SpO_2_ > 94%, without accompanying radiological signs of disease. Those with mild disease exhibited SpO_2_ > 94%, with radiological findings indicating disease presence, with or without signs of hypoxia at admission. Patients with moderately severe disease presented with severe hypoxia, fever, and multiple chest X-ray opacifications (or specific CT changes) and showed good response to oxygen therapy (SpO_2_ > 90% after receiving oxygen via nasal cannula or mask at 10–15 L/min for one hour). Patients with severe or very severe disease exhibited either initial stage or developed ARDS.

### 2.2. Data Collection, Assessment, Sample Size, and Power of Study

Data for this study were retrospectively collected from medical records and organized into a structured database. The collected information encompassed four main categories: demographic characteristics, clinical parameters, laboratory findings, and therapeutic interventions.

Demographic data included age, sex, anthropometric measurements (height and weight), and blood type. Relevant comorbidities were documented: diabetes mellitus, hypertension, cardiovascular diseases, chronic respiratory diseases, chronic kidney disease, and other significant disorders, along with information about chronic medications prior to hospitalization.

Laboratory parameters were monitored on Days 1 and 7 of hospitalization and included hematological parameters (complete blood count), inflammatory markers (C-reactive protein, ferritin, LDH), coagulation parameters (D-dimer), biochemical analyses (including lipid profile), arterial blood gas analyses (pH, pO2, pCO_2_), and clinical scores (SOFA, APACHE II, RASS).

Clinical characteristics documented at admission included disease signs and symptoms, severity of clinical presentation, and radiological findings (chest X-ray, CT, and CO-RADS scores). During hospitalization, vital signs and oxygenation parameters (oxygen saturation—SpO_2_, modified oxygenation index—SpO_2_/FiO_2_), level of consciousness (GCS, AVPU scale), and therapeutic interventions were systematically monitored, with particular attention paid to respiratory support modalities.

The minimum sample size was calculated to be 55 for the selected time period of one month during pandemic conditions and for design of the study presented (cross-sectional), considering the obtained prevalence of changed FVC (forced vital capacity) among COVID-19 patients, according to the methods of a previous study [12]. The power of the study was calculated with the alpha = 0.05, d = 0.13, and with the minimum of an 80% mean, according to the main objective of the study, using G*power software 3.1. 

### 2.3. Statistical Analysis

Statistical analyses were performed using SPSS (Statistical Package for the Social Sciences) v.23.0. Continuous variables were presented as the mean ± standard deviation or the median with interquartile range, as appropriate. Categorical variables were expressed as frequencies and percentages. Comparisons between survivors and non-survivors were performed using the Mann–Whitney U test for continuous variables and the Chi-square or Fisher’s exact test for categorical variables. Receiver operating characteristic (ROC) curves were constructed to assess the predictive value of various parameters, with the area under the curve (AUC) calculated. Statistical significance was set at *p* < 0.05.

Binary logistic regression analysis was used to examine the relationship between the dependent and independent variables. Age groupings, divided into four categories using the transform data option, were as follows: category 1 (27–41 years), category 2 (42–53 years), category 3 (54–65 years), and category 4 (above 66 years). According to the levels of BMI, all patients were divided into the non-obese category group (BMI < 30 kg/m^2^) and the obese category group (BMI > 30 kg/m^2^). The odds ratio (ORs), along with the 95% confidence interval, was used to assess the risk relative to obesity presence (BMI) and age. The statistical significance was considered to be achieved at *p* value less than 0.05, and these statistical tests were performed using IBM SPSS, version 26.0, for Macintosh.

## 3. Results

### 3.1. Demographic and Baseline Characteristics of Non-Smoking COVID-19 Patients

The mean age of participants was 57.56 ± 13.62 years (range 27–86 years). In the study cohort, males constituted the majority of patients (43/59, 72.9%) compared to number of females (16/59, 27.1%). The mean BMI was 30.15 ± 6.34 kg/m^2^ (range 17.9–45.0), placing the average patient in the obesity category (BMI ≥ 30 kg/m^2^). The mean time from symptom onset to hospitalization was 9.81 ± 5.22 days.

The overall mortality rate in the study group was 54.2% (32/59 patients). Mortality rates were 55.8% (24/43) for males and 50.0% (8/16) for females, with no statistically significant difference between sexes (*p* = 0.690).

The most common comorbidities in the study cohort were hypertension (44.1%), cardiovascular diseases (23.7%), and diabetes mellitus (18.6%). Although a trend toward higher mortality was observed in patients with comorbidities (cardiovascular diseases 64.3% vs. 51.2%; diabetes 63.6% vs. 52.2%; chronic kidney disease 66.7% vs. 54.4%), these differences did not reach statistical significance (*p* > 0.05).

### 3.2. Laboratory and Clinical Parameters

Analysis of vital and respiratory parameters on Day 1 of hospitalization revealed no significant differences between survivors and non-survivors in regards to vital signs, including heart rate (90.52 ± 17.13 vs. 89.13 ± 24.62 beats/minute; *p* = 0.364), respiratory rate (20.68 ± 6.56 vs. 21.10 ± 10.73 breaths/minute; *p* = 0.670), and temperature (36.95 ± 0.85 vs. 36.65 ± 0.54 °C; *p* = 0.271). Similarly, the analysis of respiratory parameters showed no significant differences in oxygenation indices or gas exchange parameters. Higher pO_2_ values were observed in non-survivors (90.44 ± 51.95 vs. 71.03 ± 20.05 mmHg), although without statistical significance (*p* = 0.386). No significant differences were found in pH (7.43 ± 0.09 vs. 7.40 ± 0.11; *p* = 0.468), pCO_2_ (41.73 ± 14.89 vs. 43.90 ± 16.47 mmHg; *p* = 0.796), SpO_2_ (94.03 ± 3.65 vs. 92.52 ± 5.37%; *p* = 0.456), or other related parameters such as FiO_2_, PEEP, HCO_3_, base excess, lactate levels, and oxygenation indices (SpO_2_/FiO_2_, PaO_2_/FiO_2_) (Table 1).

The analysis of inflammatory and hematological parameters showed no statistically significant differences between groups. Values of CRP (161.34 ± 99.69 vs. 145.24 ± 89.96 mg/L; *p* = 0.760), ferritin (867.00 ± 550.57 vs. 950.54 ± 467.04 µg/L; *p* = 0.606), and LDH (468.02 ± 189.46 vs. 637.36 ± 367.06 U/L; *p* = 0.218) showed no statistically significant differences. Similarly, no significant differences were found for leukocytes (15.84 ± 13.86 vs. 38.99 ± 94.41 × 10^9^/L; *p* = 0.849), lymphocytes (0.68 ± 0.28 vs. 4.75 ± 12.81 × 10^9^/L; *p* = 0.116), neutrophils (77.77 ± 256.78 vs. 155.83 ± 357.79 × 10^9^/L; *p* = 0.860), or other hematological parameters, such as hemoglobin, hematocrit, and platelets (Table 2).

Biochemical parameter analysis revealed total cholesterol as the only parameter showing a statistically significant difference between survivors and non-survivors, with higher levels in non-survivors (4.11 ± 0.94 vs. 2.89 ± 1.25 mmol/L; *p* = 0.020). Other biochemical parameters, including glucose, total protein, albumin, urea, creatinine, liver enzymes (ALT, AST), and total bilirubin, showed no significant differences between groups. Additionally, hemostatic parameters (fibrinogen, INR, D-dimer) and cardiac injury markers (hsTnI, proBNP) demonstrated no significant differences between survivors and non-survivors on Day 1 (Table 3). Moreover, analysis of electrolyte levels showed no significant differences between groups for sodium, potassium, calcium, magnesium, phosphate, or iron levels (Table 4). Clinical scores demonstrated no statistically significant differences between groups, including for SOFA (29.16 ± 120.22 vs. 29.25 ± 97.85, *p* = 0.957), RASS (45.47 ± 141.90 vs.−1.07 ± 2.73, *p* = 0.392), and APACHE II score (60.81 ± 155.49 vs. 51.32 ± 131.86, *p* = 0.813) (Table 5).

Analysis on the seventh day of hospitalization revealed several significant differences between survivors and non-survivors. Regarding vital and respiratory parameters, the pH values were significantly lower in non-surviving patients (7.34 ± 0.13 vs. 7.46 ± 0.05; *p* = 0.005), while the pCO_2_ values were significantly higher (59.70 ± 15.66 vs. 42.86 ± 8.33 mmHg; *p* = 0.004). Oxygenation parameters demonstrated significant differences, with lower SpO_2_ values (92.04 ± 5.79 vs. 96.11 ± 2.85%; *p* = 0.048) and a lower SpO_2_/FiO_2_ ratio (95.39 ± 17.16 vs. 106.64 ± 22.73; *p* = 0.032) in the non-surviving patient group. No significant differences were found in regards to vital signs, including heart rate, respiratory rate, and temperature (Table 6).

For inflammatory and hematological parameters, significantly higher values in non-surviving patients were noted for leukocytes (17.96 ± 6.81 vs. 12.63 ± 4.36 × 10^9^/L; *p* = 0.033) and neutrophils (24.16 ± 24.22 vs. 11.28 ± 4.69 × 10^9^/L; *p* = 0.025). No statistically significant differences were found regarding the following inflammatory marker values: CRP (129.43 ± 96.63 vs. 80.31 ± 71.01 mg/L; *p* = 0.166), ferritin (1162.41 ± 407.63 vs. 881.40 ± 502.44 µg/L; *p* = 0.162), and LDH (601.78 ± 318.24 vs. 481.83 ± 301.91 U/L; *p* = 0.182) (Table 7).

Biochemical parameters on Day 7, including glucose, total protein, albumin, liver enzymes, and lipid profile, showed no statistically significant differences between survivors and non-survivors. Similarly, hemostatic parameters (fibrinogen, INR, D-dimer) and cardiac injury markers (hsTnI, proBNP) demonstrated no significant differences between groups (Table 8).

Analysis of electrolyte status on Day 7 showed no significant differences between groups in sodium, potassium, calcium, magnesium, phosphate, or iron levels (Table 9). The SOFA score was significantly higher in non-surviving patients (16.86 ± 29.35 vs. 4.75 ± 3.69; *p* = 0.003), while the RASS score showed no significant difference between groups (Table 10).

### 3.3. Respiratory Support

On the first day of hospitalization, the majority of patients (80.7%) required advanced oxygen therapy, with non-invasive mechanical ventilation (NIMV) being the most frequently applied modality (52.6% of patients), followed by invasive mechanical ventilation (IMV, 28.1%) and use of a high-flow nasal cannula (HFNC, 12.3%). Only 7% of patients required basic oxygen therapy via mask.

On the seventh day of hospitalization, a significant change in the distribution of respiratory support types was observed, with a statistically significant difference between surviving and non-surviving patients (*p* = 0.012). Among ventilated patients, the highest mortality was recorded in those on IMV (84.6%), while for patients on NIMV, it was 50%. Notably, no mortality was observed in patients on HFNC on the seventh day.

To assess the predictive value of various parameters, ROC curves were constructed, and areas under the curve (AUC) were calculated. The SOFA score determined on the seventh day of hospitalization demonstrated the best discriminatory ability (AUC = 0.902; *p* = 0.004; 95% CI: 0.768–1.000). Among other seventh-day parameters, pCO_2_ showed significant predictive value (AUC = 0.853; *p* = 0.012; 95% CI: 0.696–1.000), while neutrophils exhibited borderline statistical significance (AUC = 0.775; *p* = 0.050; 95% CI: 0.557–0.992). Leukocytes displayed an AUC = 0.735 (*p* = 0.093; 95% CI: 0.492–0.979), SpO_2_ showed an AUC = 0.284 (*p* = 0.123; 95% CI: 0.051–0.518), and pH value indicated an AUC = 0.176 (*p* = 0.021; 95% CI: 0.004–0.349) (Figure 1).

### 3.4. Binary Logistic Regression Analysis of the Data

The logistic regression analysis evaluating the connection between age categories and BMI shows significantly higher odds for COVID-19 infection among obese participants in all age categories, except for the 42–53 age group, after considering the 53 years and above age category as a constant (Table 11). Thus, the odds for a COVID-19 fatal outcome are higher in obese participants in comparison those for non-obese patients, and it decreases with increasing age (Table 11).

The main risk for the negative outcome among obese participants was increasing age, according to both adjusted and unadjusted values (Table 12). However, after adjustment for age, the odds for fatal outcome among obese patients slightly increased in comparison with those for the patients with normal BMI (Table 13).

## 4. Discussion

In our study, which included 59 non-smokers with severe COVID-19, we analyzed mortality predictors at two time points. Upon admission, total cholesterol showed significant predictive value, while on the seventh day of hospitalization, multiple significant predictors emerged: SOFA score, acid–base status parameters (pH, pCO_2_), oxygenation indicators (SpO_2_, SpO_2_/FiO_2_), and hematological parameters (leukocytes, neutrophils). The high mortality rate in our cohort (54.2%) indicates the need for a deeper understanding of factors influencing COVID-19 infection severity, particularly in the context of current knowledge about smoking status and disease progression.

Although smoking is traditionally recognized as a risk factor for severe respiratory infections, the COVID-19 pandemic revealed an unexpected phenomenon, termed the “smoker’s paradox”, in which active smokers showed lower rates of hospitalization and less severe disease compared to the results for non-smokers [13,14]. While this observation has prompted numerous studies investigating the potential impact of smoking on COVID-19 infection progression, little attention has been devoted to systematic analysis of risk factors and mortality predictors specifically in the non-smoking population. Identifying reliable predictors in this understudied population can significantly contribute to a better understanding of the disease’s natural course and improve risk stratification.

In our study, males comprised the majority of participants, consistent with previous findings of higher incidence and mortality among men [15]. Higher expression of ACE2 receptors in males, along with androgen-dependent regulation of the TMPRSS2 (transmembrane serine protease), which facilitates SARS-CoV-2 virus entry into cells, potentially explain the more severe clinical outcomes. Conversely, estrogen may have a protective effect in women [16,17,18]. Although our study showed a trend towards higher mortality in males, the difference was not statistically significant, which may be a consequence of the limited sample size and the specific non-smoking population. More extensive studies indicate significantly higher mortality in the male population [19,20].

Analysis of laboratory parameters on the first day of hospitalization showed no significant differences in vital or respiratory parameters between survivors and non-survivors. However, by the seventh day of hospitalization, several significant differences emerged. Patients with adverse outcomes demonstrated significantly lower SpO_2_ values, indicating more pronounced hypoxemia and lung function impairment. These results align with those from the study conducted by Tirora et.al, which also identifies low oxygen saturation as a mortality predictor in COVID-19 patients. Pathophysiologically, this correlation can be explained by the SARS-CoV-2 virus’s specific affinity for lung tissue, where lower oxygen saturation directly reflects the degree of damage and reduced lung functional capacity [21]. Furthermore, oxygen saturation below 90% has been shown to be a strong predictor of 24 h mortality, with each 10% decrease in saturation increasing the mortality rate by approximately 2.66 times (*p* = 0.0002; 95% CI OR = 1.45–4.85) [22]. Similar findings were confirmed in a study by Marwan et al., where mortality rates were significantly higher in patients with SpO_2_ < 90% compared to those with SpO_2_ ≥ 90% [23]. SpO_2_ is the only oxygenation parameter that can be assessed without accompanying laboratory analyses, making it crucial for rapid and non-invasive oxygenation assessment in clinical practice. Its correlation with laboratory markers, such as lactate and acid–base status, further confirms its prognostic value in critically ill patients [24].

In contrast to our findings in non-smokers, studies involving smokers have demonstrated that baseline SpO_2_ values tend to be slightly lower in smokers, possibly due to chronic adaptations to lower oxygen levels. A study reported that 21.2% of smokers with COVID-19 had SpO_2_ levels below 75%, indicating a strong effect of smoking on oxygen desaturation. In smokers, elevated carboxyhemoglobin levels may also contribute to this reduction in oxygen saturation [25].

Our study results showed that the SpO_2_/FiO_2_ ratio on the seventh day was statistically significantly lower in the deceased patient group, consistent with a recent study by Zinna et al. that indicated an independent association between the SpO_2_/FiO_2_ ratio and in-hospital mortality in COVID-19 patients [26]. Similarly, it has been demonstrated that the SpO_2_/FiO_2_ ratio measured on the 2nd and 3rd days of hospitalization is independently associated with COVID-19 mortality, with predictive value for 28-day mortality [27]. While the PaO_2_/FiO_2_ ratio is the gold standard in diagnosing acute respiratory insufficiency, the SpO_2_/FiO_2_ ratio strongly correlates with PaO_2_/FiO_2_ in patients with COVID-19 pneumonia, enabling early therapy adjustment and improving overall patient survival [26,28,29].

Moreover, analysis of arterial blood gas analyses on the seventh day revealed that patients with adverse outcomes displayed significantly higher partial carbon dioxide pressure (pCO_2_) values, implying respiratory insufficiency. This finding supports the results of Koc et al.’s study, which identified pCO_2_ as an independent mortality predictor in COVID-19 patients with acute respiratory failure [30]. In the context of COVID-19 infection, hypercapnia may result from multiple pathological mechanisms, including severe ARDS, respiratory muscle fatigue, and inadequate response to mechanical ventilation [31,32].

Severe COVID-19 disease is characterized by lung parenchymal damage, which leads to acidosis, as reflected in the significantly lower pH values we observed in non-survivors. Hypoxia can influence ACE2 receptor expression regulation, while increased lactate levels affect intra- and extracellular pH, potentially facilitating SARS-CoV-2 entry into host cells and disease progression [33,34]. These findings align with those in previous research showing that acid–base imbalance, particularly combined respiratory and metabolic acidosis, is associated with increased mortality risk [35]. Therefore, continuous monitoring and timely correction of acid–base status can play a crucial role in improving patient outcomes.

The inflammatory response plays a central role in COVID-19 progression. SARS-CoV-2 initiates a cascade of inflammatory processes that can lead to a cytokine storm, a primary factor in severe clinical outcomes [36]. In our study, although CRP values upon admission were higher in deceased patients, the difference was not statistically significant, which aligns with findings from Davoudi et al. [37]. Similarly, parameters such as ferritin, LDH, and D-dimer levels showed numerical, but non-significant, differences upon admission. These findings warrant special attention in the context of our sample size, given that the significance of these parameters in predicting COVID-19 infection outcomes has been confirmed in larger studies. A meta-analysis by Huang et al. demonstrated that elevated CRP, D-dimer, and ferritin levels are associated with poor disease outcomes [38]. Ferritin can contribute to cytokine storm development through direct immunomodulatory effects, while elevated D-dimer values indicate coagulation disorders, a significant COVID-19 complication [39]. LDH serves as a sensitive marker of tissue damage, potentially reflecting the degree of damage caused by COVID-19 infection [40].

In contrast to our findings in non-smoker patients, studies including smokers have demonstrated higher baseline levels of inflammatory biomarkers. Research comparing healthy smokers and non-smokers found significantly elevated D-dimer and fibrinogen levels in smokers [41]. During COVID-19 infection, these pre-existing elevated markers may complicate the interpretation of disease severity in smokers. Therefore, our observations in non-smokers represent inflammatory responses directly attributable to COVID-19 severity, without the confounding effect of smoking-induced chronic inflammation.

Hematological parameters, including hemoglobin, leukocytes, and platelets, are important for monitoring disease progression, and their dynamics may reflect disease severity. Leukocytes, especially neutrophils, play a crucial role in immune defense and inflammation [42]. However, their excessive activation can lead to an exaggerated inflammatory reaction, tissue damage, and disease progression, significantly contributing to the development of inflammatory and hemorrhagic lesions in SARS-CoV infection [43].

On Day 7, our analysis showed significantly elevated leukocyte and neutrophil counts in non-survivors, suggesting a stronger systemic inflammatory response. This finding aligns with those of previous studies linking leukocytosis and neutrophilia with higher mortality in COVID-19 patients [44,45]. However, Liu et al. documented that 80% of their patients showed normal or reduced leukocyte counts [46], similar to other reports showing that leukocytosis is not a universal characteristic of severe COVID-19 forms [47]. Thus, while often predictive of poor outcomes, leukocytosis is not universally present, reflecting the complexity of the immune response to COVID-19.

At the beginning of the pandemic, it was observed that patients with severe clinical disease progression often displayed reduced levels of circulating lymphocytes. Early data from Wuhan in 2020 showed markedly lower lymphocyte percentages in deceased patients (<5%) compared to the levels in survivors (>20%) [48,49]. Although our study recorded lower lymphocyte values in the deceased patient group, this difference did not reach statistical significance. This may reflect inter-individual variability, timing of sampling, or differences in lymphocytopenia dynamics throughout the disease course.

Analysis of Day 1 laboratory parameters revealed significantly higher total cholesterol levels in deceased patients, consistent with studies linking lipid profile to COVID-19 infection severity [50]. Cholesterol may facilitate viral fusion with the host cell membrane via lipid rafts, increasing cell susceptibility to SARS-CoV-2 infection [51]. The biochemical parameters, specifically for lipids, did not exhibit significant differences on Day 7. Other studies have reported changes in lipid profiles, particularly cholesterol and triglycerides, during the progression of severe COVID-19.

Analysis of the seventh-day hospitalization parameters highlighted the SOFA score as a significant mortality predictor. The SOFA score, a validated instrument for evaluating the degree of organ dysfunction and outcomes in critically ill patients, has demonstrated predictive value across various clinical conditions, including hematological malignancies and chronic liver insufficiency [52]. Its relevance in COVID-19 is well established, as SARS-CoV-2 affects multiple organ systems beyond the lungs, including the cardiovascular, hepatic, renal, nervous, and endocrine systems [53]. The fact that the SOFA score demonstrates significant predictive value on the seventh day indicates the necessity of dynamically monitoring this parameter during hospitalization. However, despite its widespread use, some studies have shown that patient age was a superior mortality predictor for COVID-19 patients compared to the SOFA score [54]. This discrepancy underscores the need for caution when applying existing predictive models to novel diseases, especially for making therapeutic decisions.

Additionally, prognostic scores such as SOFA may display different predictive values in the smoking population. The chronic systemic inflammation and altered baseline organ function in smokers could potentially modify the interpretation and prognostic accuracy of such scores, highlighting the importance of separate validations in distinct patient populations. However, to date, there are no studies specifically comparing SOFA score values between smokers and non-smokers infected with COVID-19, warranting further research to address this gap.

Analysis of applied respiratory support revealed significant dynamics during hospitalization. Unlike classic ARDS, where deterioration of gas exchange is accompanied by decreased pulmonary compliance, COVID-19 patients exhibit a unique pattern, with the simultaneous presence of hyperperfused and hypoperfused regions in lung parenchyma. This specific pathology leads to ventilation–perfusion (V/Q) mismatch, which can result in hypoxemia, even with normal or increased static compliance. Such a pathophysiological mechanism makes selecting the optimal type of respiratory support particularly challenging [55]. In our study, while most patients were on non-invasive mechanical ventilation (NIMV) on the first day, by the seventh day, there was a significant change in distribution, with invasive mechanical ventilation (IMV) becoming dominant. Notably, mortality was the highest in patients on IMV, significantly lower in patients on NIMV, and no deaths were recorded in patients on a high-flow nasal cannula (HFNC) on the seventh day. This finding aligns with a growing body of evidence pointing to potential HFNC advantages. The literature suggests that early HFNC application can prevent intubation in a significant number of COVID-19 pneumonia patients, with studies showing that up to one-third of cases may avoid the need for invasive ventilation. However, significant variability in non-invasive respiratory support failure rates and difficulties in predicting which patients will require intubation make the decision to escalate respiratory support particularly challenging [56].

Of note, a large meta-analysis revealed that smoking, while associated with increased disease severity and mortality, did not correlate with an increased need for mechanical ventilation. This paradox may reflect pre-existing adaptations to hypoxemia in smokers or nicotine’s acute effects on inflammatory pathways [57].

The mortality rate among patients on IMV in our study was higher compared to the results of a large meta-analysis that included 12,437 COVID-19 patients from intensive care units, where mortality in patients on IMV was 43% (95% CI 0.29–0.58). The meta-analysis confirmed that IMV application is a strong mortality predictor, especially when combined with acute kidney failure and ARDS [58]. Regional differences in outcomes suggest the need for developing respiratory support protocols tailored to the specificities of local healthcare systems. Our results showing lower mortality in patients on NIMV and HFNC emphasize the importance of optimizing ventilation strategies and carefully selecting the timing for respiratory support escalation.

In the context of our study, which exclusively included non-smokers with severe COVID-19 clinical presentation, special attention is paid to the consideration of the so-called “smoker’s paradox”. This phenomenon, first observed during the early phases of the pandemic, presents an intriguing contrast to the well-documented harmful effects of smoking on the respiratory system.

Smoking is a recognized risk factor for developing numerous cardiometabolic and respiratory diseases, including chronic obstructive pulmonary disease and bronchial asthma. Tobacco consumption leads to exposure to numerous toxic chemicals, such as 1,3-butadiene, benzene, and NO_2_, which cause respiratory tract inflammation and allergic reactions, increase epithelial cell permeability, stimulate mucus formation, and disrupt mucociliary transport [59]. Consequently, active smokers more frequently contract respiratory infections like influenza and MERS [60].

However, epidemiological data collected during the COVID-19 pandemic showed unexpectedly lower hospitalization rates and less severe disease forms among active smokers compared to non-smokers [13,14,61].

Several biological mechanisms potentially explain this paradox, with the most significant being smoking’s influence on ACE2 receptor expression in the respiratory tract. Given that the ACE2 receptor is the primary binding site for SARS-CoV-2, tobacco smoke exposure can significantly impact infection risk. Interestingly, while nicotine may induce ACE2 receptor expression in the lower respiratory tracts, this increased expression can paradoxically have a protective effect [62]. ACE2 possesses anti-inflammatory and antioxidative properties that may protect lung tissue from excessive immune response damage. Additionally, increased concentrations of soluble ACE2 in smokers’ serum could potentially neutralize the virus before it comes into contact with cellular receptors [14,63].

Beyond its impact on ACE2, nicotine may have broader immunomodulatory effects through the activation of the cholinergic anti-inflammatory pathway. This neuroimmune mechanism, mediated by the vagus nerve, involves α7-nicotinic acetylcholine receptors (α7-nAChR), present in both the central nervous system and immune cells. Activation of these receptors suppresses pro-inflammatory cytokines, including TNF-α, IL-1β, IL-6, and IL-17A, potentially limiting an excessive inflammatory response in the lungs [14,64]. Additionally, research has shown that acute exposure to tobacco smoke results in increased NO concentration in the lower respiratory tracts, with NO bioequivalents acting protectively against SARS-CoV-2 aerosol particles [11].

Despite these potentially protective mechanisms, clinical trials of nicotine therapy have not yielded encouraging results. Randomized studies unequivocally showed that transdermal nicotine application did not significantly improve outcomes in patients with severe COVID-19 pneumonia. Despite intriguing findings about the potential protective effects of certain tobacco smoke components, the harmful effects of smoking on overall health remain indisputable [65,66].

In light of these findings, our research, focused on non-smokers, gains additional significance. While numerous studies have investigated how smoking affects COVID-19 outcomes, less attention has been devoted to patients who have never smoked. The absence of potentially protective mechanisms associated with smoking raises the question of which other factors may be crucial in determining outcomes in this population. Identifying reliable mortality predictors in non-smokers can significantly contribute to more precise risk stratification and improved therapeutic approaches.

Although significant progress has been made in the development of therapeutic strategies for COVID-19, including anti-inflammatory drugs, angiotensin-converting enzyme inhibitors/angiotensin receptor blockers, nucleoside analogues, protease inhibitors, and monoclonal antibodies, clinical management of severe cases remains challenging. Supportive therapies like vitamins D and B might help modulate the immune response in viral infections [67,68], but accurate risk stratification based on clinical and biochemical predictors remains essential.

Regarding the inflammatory and hematological parameters, many others have reported the relevance of emergent systemic inflammation indices (NHL, NLR, RDW, SII, and SIRI, among others), for predicting severe COVID-19, IMV support, and low survival probability during hospitalization due to COVID-19 in patients when compared to the power of individual clinical markers. Recent research has highlighted the superior prognostic value of composite inflammatory indices over individual biomarkers in COVID-19, prompting a shift toward integrated hematological ratios that better reflect the systemic immune response. Markers such as the neutrophil-to-lymphocyte ratio (NLR), derived NLR (dNLR), platelet-to-lymphocyte ratio (PLR), monocyte-to-lymphocyte ratio (MLR), red cell distribution width (RDW), and neutrophil-to-hemoglobin and lymphocyte ratio (NHL) have consistently been associated with adverse clinical outcomes [69,70]. Beyond these, more complex indices—including the systemic immune-inflammation index (SII), systemic inflammation response index (SIRI), and aggregate index of systemic inflammation (AISI)—offer additional insight into inflammatory status and disease prognosis [71]. These markers have demonstrated clinical relevance across different patient populations and care settings [72,73]. Their integration into future studies on non-smoking COVID-19 patients could enhance early risk stratification and help identify those at higher risk for poor outcomes.

Our findings reinforce the need for individualized patient assessment, particularly in non-smokers, where distinct pathophysiological mechanisms may contribute to disease severity.

Our study has several limitations that must be acknowledged. First, the small sample size limits statistical power, particularly when analyzing multiple variables. Second, the retrospective design introduces the risk of selection bias and limits our ability to control for all potential confounding factors. Third, the study was conducted at a single center during a specific phase of the pandemic, which may restrict the generalizability of our findings to other healthcare settings or viral variants. Fourth, certain laboratory parameters, particularly the lipid profile, were unavailable for all patients, limiting our ability to conclude specific biomarkers. Fifth, we focused on individual inflammatory markers rather than composite inflammatory indices, which have demonstrated predictive value in recent studies. Finally, information about other pollutant exposures, mainly biomass-burning smoke, which has been clearly described as an environmental worsening factor in COVID-19 patients, were not included. These limitations underscore the need for larger, prospective, multicenter studies to validate our findings and further investigate mortality predictors in non-smoking COVID-19 patients.

## 5. Conclusions

In this research, we endeavored to identify early mortality predictors in non-smoking patients with severe COVID-19 by evaluating clinical, laboratory, and oxygenation parameters. An unexpected pattern, known as the “smoker’s paradox,” has been identified, with lower rates of severe disease among smokers compared to non-smokers, highlighting the need for the specific investigation of disease progression in non-smoking populations.

Our study identified several key mortality predictors in non-smokers with severe COVID-19, with the seventh-day SOFA score emerging as the strongest predictor. Significant differences in gas exchange parameters and hematological markers on the seventh day, along with varying mortality rates, depending on respiratory support type, demonstrate the complex nature of disease progression in this population. While research into the “smoker’s paradox” continues, our findings emphasize the importance of identifying specific risk factors in non-smokers through the timely monitoring of clinical parameters. Future research should expand on these findings, employing larger cohorts to develop more precise predictive models for this specific patient group.

## Figures and Tables

**Figure 1 healthcare-13-01041-f001:**
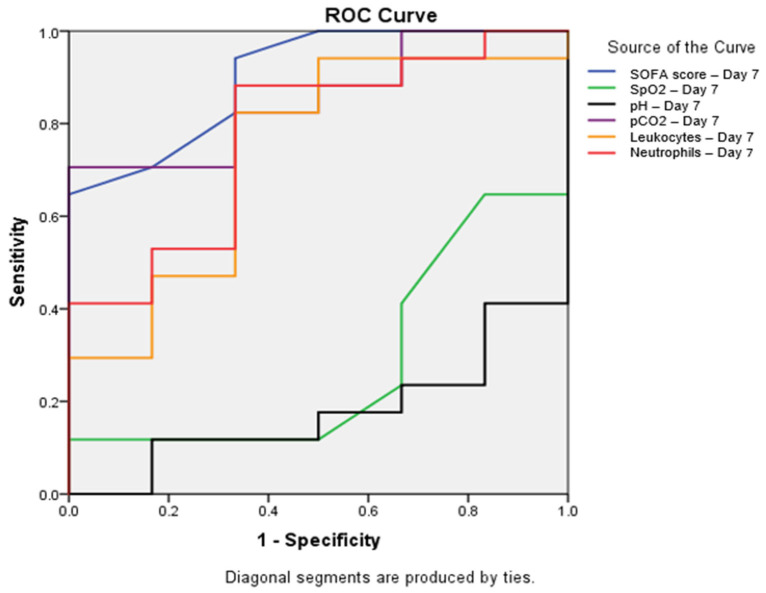
Receiver operating characteristic (ROC) curve for parameters with significant predictive value on Day 7. ROC curves showing the predictive value of the SOFA score (blue line, AUC = 0.902, *p* = 0.004), SpO_2_ (green line, AUC = 0.284, *p* = 0.123), pH (black line, AUC = 0.176, *p* = 0.021), pCO_2_ (purple line, AUC = 0.853, *p* = 0.012), leukocytes (orange line, AUC = 0.735, *p* = 0.093), and neutrophils (red line, AUC = 0.775, *p* = 0.050) on Day 7 for mortality prediction. The SOFA score showed the highest predictive value, with AUC = 0.902 (95% CI: 0.768–1.000), followed by pCO_2_, with AUC = 0.853 (95% CI: 0.696–1.000). All parameters shown demonstrated statistically significant differences between survivors and non-survivors (*p* < 0.05). Abbreviations: SOFA—sequential organ failure assessment; SpO_2_—peripheral oxygen saturation; pCO_2_—partial pressure of carbon dioxide; ROC—receiver operating characteristic.

**Table 1 healthcare-13-01041-t001:** Vital and respiratory parameters—Day 1.

Parameter	Total			Non-Survivor			Survivor			*p*
	N	Mean	SD	N	Mean	SD	N	Mean	SD	
Heart rate (beats/minute)	55	89.91	20.537	31	90.52	17.133	24	89.13	24.621	0.364
Respiratory rate (breaths/minute)	45	20.87	8.556	25	20.68	6.556	20	21.1	10.731	0.670
Temperature (°C)	55	36.815	0.7405	31	36.945	0.8512	24	36.646	0.5381	0.271
FiO_2_ (%)	52	93.46	24	30	95.33	19.429	22	90.91	29.424	0.718
PEEP (cmH_2_O)	37	8.86	2.123	22	8.64	2.46	15	9.2	1.521	0.356
SpO_2_ (%)	56	93.36	4.518	31	94.03	3.647	25	92.52	5.371	0.456
pH	52	7.4167	0.10035	28	7.4307	0.08888	24	7.4005	0.112	0.468
pO_2_ (mmHg)	53	81.653	41.5501	29	90.441	51.9533	24	71.033	20.0548	0.386
pCO_2_ (mmHg)	53	42.717	15.5122	29	41.734	14.8946	24	43.904	16.4701	0.796
HCO_3_ (mmol/L)	51	26.447	4.5344	28	27.261	5.2812	23	25.457	3.2603	0.426
BE (mmol/L)	51	3.012	6.2924	28	3.961	7.5189	23	1.857	4.25	0.691
SpO_2_/FiO_2_ ratio	53	102.58	45.282	29	96.33	13.705	24	110.14	65.562	0.726
PaO_2_/FiO_2_ ratio	53	84.3491	50.4242	29	92.8276	55.44564	24	74.1042	42.49471	0.520
Lactate (mmol/L)	52	1.423	1.1442	29	1.517	1.4296	23	1.304	0.6385	0.985

Abbreviations: FiO_2_—fraction of inspired oxygen; PEEP—positive end-expiratory pressure; SpO_2_—peripheral oxygen saturation; pO_2_—partial pressure of oxygen; pCO_2_—partial pressure of carbon dioxide; HCO_3_—bicarbonate; BE—base excess.

**Table 2 healthcare-13-01041-t002:** Inflammatory and hematological parameters—Day 1.

Parameter	Total			Non-Survivor			Survivor			*p*
	N	Mean	SD	N	Mean	SD	N	Mean	SD	
Leukocytes (×10^9^/L)	51	25.824	63.1278	29	15.838	13.8612	22	38.986	94.4087	0.849
Lymphocytes (×10^9^/L)	50	2.388	8.43729	29	0.6776	0.27874	21	4.75	12.81412	0.116
Neutrophils (×10^9^/L)	50	110.5568	302.3966	29	77.7707	256.78475	21	155.8329	357.78805	0.860
Hb (g/L)	51	125.76	19.975	29	125.62	19.001	22	125.95	21.645	0.909
HCT (L/L)	51	0.3823	0.06569	29	0.3795	0.06356	22	0.3861	0.06972	0.717
Platelets (×10^9^/L)	51	247.27	95.832	29	240.31	74.939	22	256.45	119.261	0.924
CRP (mg/L)	49	154.769	95.2079	29	161.341	99.6909	20	145.24	89.9582	0.760
PCT (ng/mL)	40	0.8865	2.33927	24	0.5417	0.84278	16	1.4037	3.5585	0.300
LDH (U/L)	36	533.87	280.338	22	468.02	189.455	14	637.36	367.06	0.218
Ferritin (µg/mL)	36	897.17	516.7	23	867	550.567	13	950.54	467.044	0.606

Abbreviations: Hb—hemoglobin; HCT—hematocrit; CRP—C-reactive protein; PCT—procalcitonin; LDH—lactate dehydrogenase.

**Table 3 healthcare-13-01041-t003:** Biochemical parameters—Day 1.

Parameter	Total			Non-Survivor			Survivor			*p*
	N	Mean	SD	N	Mean	SD	N	Mean	SD	
Glucose (mmol/L)	49	9.108	5.1239	29	9.786	5.8457	20	8.125	3.7746	0.304
Total protein (g/L)	37	58.24	12.722	23	60.22	8.431	14	55	17.598	0.684
Albumin (g/L)	40	24.23	6.171	24	24.92	5.124	16	23.19	7.539	0.719
Urea (mmol/L)	49	9.184	13.4173	28	7.3	3.5983	21	11.695	20.0746	0.585
Creatinine (µmol/L)	50	87.96	34.618	29	87.17	38.776	21	89.05	28.791	0.783
ALT (U/L)	45	73.31	51.266	25	77.96	49.699	20	67.5	53.874	0.244
AST (U/L)	45	69.22	45.057	25	68.96	51.168	20	69.55	37.337	0.552
Total bilirubin (µmol/L)	29	9.114	4.2165	18	9.039	3.7944	11	9.236	5.0278	0.822
Lipid profile										
Total cholesterol (mmol/L)	19	3.6579	1.19644	12	4.1083	0.94336	7	2.8857	1.24957	0.020 *
HDL cholesterol (mmol/L)	2	0.74	0.49497	1	1.09	/	1	0.39	/	0.317
LDL cholesterol (mmol/L)	3	1.4	1.77764	2	1.7	2.40416	1	0.8	/	1.000
Triglycerides (mmol/L)	20	1.815	0.86101	13	1.9231	0.71549	7	1.6143	1.11867	0.131
Hemostatic parameters										
Fibrinogen (g/L)	19	3.9211	1.03847	10	4.21	1.27318	9	3.6	0.61847	0.306
INR	18	1.2672	0.24781	11	1.2418	0.23116	7	1.3071	0.28617	0.751
D dimer (mg/L FEU)	22	5.895	9.98104	14	4.9807	9.25312	8	7.495	11.62999	0.432
Cardiac injury parameters										
hsTnI (ng/mL)	26	147.0769	451.8537	15	188.533	586.8363	11	90.5455	149.02507	0.512
ProBNP (pg/mL)	23	2497.522	3415.367	12	1735.8333	3186.85775	11	3328.4545	3610.40855	0.157

* Statistically significant difference (*p* < 0.05). Abbreviations: ALT—alanine aminotransferase; AST—aspartate aminotransferase; HDL—high-density lipoprotein; LDL—low-density lipoprotein; INR—international normalized ratio; hsTnI—high-sensitivity troponin I; ProBNP—pro B type natriuretic peptide.

**Table 4 healthcare-13-01041-t004:** Electrolytes—Day 1.

Parameter	Total			Non-Survivor			Survivor			*p*
	N	Mean	SD	N	Mean	SD	N	Mean	SD	
Sodium (mmol/L)	50	140.84	5.128	29	140.69	5.85	21	141.05	4.056	0.700
Potassium (mmol/L)	50	5.154	6.2247	29	4.29	0.7427	21	6.348	9.5697	0.798
Calcium (mmol/L)	35	1.806	0.41	22	1.744	0.43	13	1.911	0.3658	0.208
Magnesium (mmol/L)	35	0.866	0.1714	21	0.881	0.1834	14	0.843	0.1555	0.596
Phosphate (mmol/L)	29	1.141	0.3179	18	1.161	0.3483	11	1.109	0.2737	0.928
Iron (µmol/L)	25	6.68	4.891	16	5.56	3.741	9	8.66	6.216	0.459

**Table 5 healthcare-13-01041-t005:** Clinical severity scores—Day 1.

Parameter	Total			Non-Survivor			Survivor			*p*
	N	Mean	SD	N	Mean	SD	N	Mean	SD	
SOFA score	41	29.2	110.732	25	29.16	120.215	16	29.25	97.851	0.957
RASS score	44	30.66	118.584	30	45.47	141.897	14	−1.07	2.731	0.392
APPACHE II score	42	57.42	145.91	27	60.81	155.489	15	51.32	131.859	0.813

Abbreviations: SOFA—sequential organ failure assessment; RASS—Richmond Agitation–Sedation Scale.

**Table 6 healthcare-13-01041-t006:** Vital and respiratory parameters—Day 7.

Parameter	Total			Non-Survivor			Survivor			*p*
	N	Mean	SD	N	Mean	SD	N	Mean	SD	
Heart rate (beats/minute)	34	100.588	20.77603	25	103.8	20.3695	9	91.6667	20.34699	0.089
Respiratory rate (breaths/minute)	26	18.1538	4.23029	19	17.6842	4.0559	7	19.4286	4.75595	0.267
Temperature (°C)	34	36.6	3.48416	25	36.344	4.03537	9	37.3111	0.66978	0.411
FiO_2_ (%)	34	94.7059	11.34455	25	96	10.40833	9	91.1111	13.64225	0.115
PEEP (cmH_2_O)	30	10.2	2.13993	23	10.5652	2.06323	7	9	2.08167	0.066
SpO_2_ (%)	34	93.1176	5.44818	25	92.04	5.79137	9	96.1111	2.848	0.048 *
pH	34	7.37	0.12413	25	7.3382	0.12822	9	7.4582	0.04822	0.005 *
pO_2_ (mmHg)	34	78.4618	39.9588	25	73.66	39.61119	9	91.8	40.08828	0.076
pCO_2_ (mmHg)	34	55.2441	15.87589	25	59.704	15.65675	9	42.8556	8.33428	0.004 *
HCO_3_ (mmol/L)	34	28.9412	5.10657	25	28.58	5.23752	9	29.9444	4.87163	0.770
BE (mmol/L)	34	6.5735	7.21603	25	5.684	6.34768	9	9.0444	9.19635	0.626
SpO_2_/FiO_2_ ratio	34	98.3676	19.10211	25	95.388	17.16451	9	106.6444	22.72664	0.032 *
PaO_2_/FiO_2_ ratio	34	85.5382	54.79518	25	79.88	58.04392	9	101.2556	43.59023	0.058
Lactate (mmol/L)	34	1.7353	0.91217	25	1.88	0.94736	9	1.3333	0.70178	0.092

* Statistically significant difference (*p* < 0.05). Abbreviations: FiO_2_—fraction of inspired oxygen; PEEP—positive end-expiratory pressure; SpO_2_—peripheral oxygen saturation; pO_2_—partial pressure of oxygen; pCO_2_—partial pressure of carbon dioxide; HCO_3_—bicarbonate; BE—base excess.

**Table 7 healthcare-13-01041-t007:** Inflammatory and hematological parameters—Day 7.

Parameter	Total			Non-Survivor			Survivor			*p*
	N	Mean	SD	N	Mean	SD	N	Mean	SD	
Leukocytes (×10^9^/L)	34	16.55	6.63815	25	17.96	6.81426	9	12.6333	4.35833	0.033 *
Lymphocytes (×10^9^/L)	25	0.9324	1.03231	18	0.7528	0.66507	7	1.3943	1.63209	0.193
Neutrophils (×10^9^/L)	25	20.5536	21.35107	18	24.1594	24.22068	7	11.2814	4.69198	0.025 *
Hb (g/L)	34	114.529	21.70775	25	115.56	22.7506	9	111.6667	19.45508	0.845
HCT (L/L)	34	0.3588	0.07155	25	0.362	0.07455	9	0.35	0.06576	0.922
Platelets (×10^9^/L)	34	274.088	104.9275	25	259.88	86.036	9	313.5556	144.31053	0.532
CRP (mg/L)	34	116.429	92.18049	25	129.432	96.6311	9	80.3111	71.01043	0.166
PCT (ng/mL)	31	3.2503	5.40664	25	3.352	5.42435	6	2.8267	5.82151	0.293
LDH (U/L)	24	571.792	312.22664	18	601.7778	318.2373	6	481.8333	301.91086	0.182
Ferritin (µg/mL)	27	1110.37	430.60612	22	1162.409	407.6336	5	881.4	502.43935	0.162

* Statistically significant difference (*p* < 0.05). Abbreviations: Hb—hemoglobin; HCT—hematocrit; CRP—C-reactive protein; PCT—procalcitonin; LDH—lactate dehydrogenase.

**Table 8 healthcare-13-01041-t008:** Biochemical parameters—Day 7.

Parameter	Total			Non-Survivor			Survivor			*p*
	N	Mean	SD	N	Mean	SD	N	Mean	SD	
Glucose (mmol/L)	32	9.3031	5.49108	23	10.0826	6.12117	9	7.3111	2.75474	0.090
Total protein (g/L)	28	55.3214	5.06296	21	55.2857	4.89022	7	55.4286	5.96817	0.650
Albumin (g/L)	32	24.0625	4.36952	24	23.5	4.24264	8	25.75	4.59036	0.197
Urea (mmol/L)	33	17.4121	14.9704	24	19.1583	16.50989	9	12.7556	8.9024	0.627
Creatinine (µmol/L)	33	147.976	181.83721	24	170.8833	208.5571	9	86.8889	37.88946	0.385
ALT (U/L)	31	80.2903	73.04391	24	84.5833	80.6538	7	65.5714	37.61142	0.925
AST (U/L)	31	53.129	38.68397	24	57.2917	42.63748	7	38.8571	14.36928	0.298
Total bilirubin (µmol/L)	22	23.65	21.05163	17	25.0882	23.28532	5	18.76	10.91526	0.906
Lipid profile										
Total cholesterol (mmol/L)	16	3.3375	1.06011	12	3.1583	1.11474	4	3.875	0.74106	0.163
HDL cholesterol (mmol/L)	1	0	/	1	0	/	/	/	/	/
LDL cholesterol (mmol/L)	1	0	/	1	0	/	/	/	/	/
Triglycerides (mmol/L)	16	2.825	1.68226	12	2.6833	1.70658	4	3.25	1.77482	0.467
Hemostatic parameters										
Fibrinogen (g/L)	18	3.2722	1.08183	14	3.1429	0.96454	4	3.725	1.49972	0.456
INR	9	1.5756	1.05622	6	1.0933	0.56383	3	2.54	1.25491	0.302
D dimer (mg/L FEU)	5	3.186	2.62657	4	3.5475	2.8857	1	1.74	/	0.480
Cardiac injury parameters										
hsTnI (ng/mL)	23	215.217	674.26276	17	282.1765	778.7685	6	25.5	34.65112	0.188
ProBNP (pg/mL)	26	4532.89	9378.7004	20	5128.66	10441.9	6	2547	4395.62023	0.808

Abbreviations: ALT—alanine aminotransferase; AST—aspartate aminotransferase; HDL—high-density lipoprotein; LDL—low-density lipoprotein; INR—international normalized ratio; hsTnI—high-sensitivity troponin I; ProBNP—pro B type natriuretic peptide.

**Table 9 healthcare-13-01041-t009:** Electrolytes—Day 7.

Parameter	Total			Non-Survivor			Survivor			*p*
	N	Mean	SD	N	Mean	SD	N	Mean	SD	
Sodium (mmol/L)	33	142.727	4.51576	24	143.2083	4.84525	9	141.4444	3.39526	0.405
Potassium (mmol/L)	33	6.6121	9.7983	24	7.35	11.46192	9	4.6444	0.57687	0.430
Calcium (mmol/L)	33	1.9212	0.3712	24	1.8583	0.405	9	2.0889	0.19003	0.094
Magnesium (mmol/L)	33	1.0061	0.1731	24	1.0208	0.1744	9	0.9667	0.17321	0.606
Phosphate (mmol/L)	33	1.3788	0.98734	24	1.5042	1.13194	9	1.0444	0.20683	0.393
Iron (µmol/L)	20	9	7.98683	15	8.8667	8.77388	5	9.4	5.77062	0.457

**Table 10 healthcare-13-01041-t010:** Clinical severity scores—Day 7.

Parameter	Total			Non-survivor			Survivor			*p*
	N	Mean	SD	N	Mean	SD	N	Mean	SD	
SOFA score	30	13.6333	25.62661	22	16.8636	29.34896	8	4.75	3.69362	0.003 *
RASS score	29	57.0345	181.80985	23	72.3913	202.1799	6	−1.833	2.48328	0.226

* Statistically significant difference (*p* < 0.05). Abbreviations: SOFA—sequential organ failure assessment; RASS—Richmond Agitation–Sedation Scale.

**Table 11 healthcare-13-01041-t011:** Binary logistic regression model for age-categories as the predictor for outcome.

Age Categories	Coefficient	*p*-Value	OR	95% CI for OR	
				Lower	Upper
27–41 years	0.344	<0.000	1.652	1.543	1.798
42–53 years	0.254	0.069	1.366	1.233	1.455
54–65 years	0.020	<0.000	1.040	0.997	1.109
Above 66 years	−0.113	<0.000	0.788	0.677	0.890
Constant	−0.483	<0.000	0.498	---	---

**Table 12 healthcare-13-01041-t012:** Adjusted binary logistic regression model for age and obesity categories as predictors of outcome.

Categories	Coefficient	*p*-Value	OR	95% CI for OR	
				Lower	Upper
Obesity (Yes/No)	0.678	<0.000	1.677	1.568	1.714
27–41 years	3.109	<0.000	56.661	34.331	120.543
42–53 years	3.409	<0.000	76.023	51.876	134.786
54–65 years	3.225	<0.000	22.031	18.23	39.08
Above 66 years	1.900	<0.000	3.78	2.541	2.689
Constant	−3.207	<0.000	0.021	---	---

**Table 13 healthcare-13-01041-t013:** Adjusted and unadjusted odds of outcome for age (in years) and obesity (yes/no).

	Adjusted OR (95% CI) *	Unadjusted OR (95%)
Age (years)	0.876 (0.855–0.891)	0.875 (0.855–0.891)
Obese	0.462 (0.423–0.521)	0.439 (0.419–0.554)
Non-Obese	1.643 (1.453–2.011)	1.756 (1.501–2.153)

* adjusted for age and obesity category.

## Data Availability

The data presented in this study are available on request from the corresponding author.

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
