# Peer review of "Severe COVID-19 in Non-Smokers: Predictive Factors and Outcomes"

_healthcare, 2025, doi:10.3390/healthcare13091041_

Round 1
Reviewer 1 Report
Comments and Suggestions for Authors
The authors present valuable insights in their manuscript titled Severe COVID-19 in Non-Smokers: Predictive Factors and Outcomes. However, several revisions are required to enhance the clarity and precision of the manuscript.
- Line 41: Provide specific examples of past epidemics to improve readability and context.
- Line 46: Specify the timeline for the reported 7 million deaths (e.g., "7 million reported deaths worldwide as of [month, year]").
- Throughout the manuscript: Standardize the notation for pOâ‚‚, pCOâ‚‚, etc., ensuring consistency in formatting. It wont be pO2 or pCO2 etc.
- Line 119: Specify the version of SPSS used for data analysis.
- Tables: Highlight statistically significant parameters for better readability.
- Discussion Section: Compare the observed parameters for smokers in previous studies with the obtained values for non-smokers to provide a clearer contrast.
Author Response
Comment 1: The authors present valuable insights in their manuscript titled Severe COVID-19 in Non-Smokers: Predictive Factors and Outcomes. However, several revisions are required to enhance the clarity and precision of the manuscript. Line 41: Provide specific examples of past epidemics to improve readability and context.
Response 1: Thank you for your suggestion. We have added specific examples of previous epidemics in the 21st century to provide better context.
Comment 2: Line 46: Specify the timeline for the reported 7 million deaths (e.g., "7 million reported deaths worldwide as of [month, year]").
Response 2: Thank you for your suggestion. We made corrections.
Comment 3: Throughout the manuscript: Standardize the notation for pOâ‚‚, pCOâ‚‚, etc., ensuring consistency in formatting. It wont be pO2 or pCO2 etc.
Response 3: Thank you for your suggestion. We made corrections.
Comment 4: Line 119: Specify the version of SPSS used for data analysis.
Response 4: Thank you for your suggestion. We made a correction.
Comment 5: Tables: Highlight statistically significant parameters for better readability.
Response 5: Thank you for your suggestion. We made a correction.
Comment 6: Discussion Section: Compare the observed parameters for smokers in previous studies with the obtained values for non-smokers to provide a clearer contrast.
Response 6: Thank you for your suggestion. We made a correction.
Reviewer 2 Report
Comments and Suggestions for Authors
Thanks for the opportunity to review the manuscript by Djuric and colleagues. In this research, the authors tried to identify early mortality predictors in non-smoking patients with severe COVID-19 by evaluating clinical, laboratory, and oxygenation parameters.
The authors argue an unexpected pattern known as the "smoker's paradox," with lower rates of severe disease among smokers compared to non-smokers, highlighting the need for specific investigation of disease progression in non-smoking populations.
The main idea is interesting, but before I make a final recommendation about its acceptance, I have some concerns for the authors that must be clarified.
The authors included only 59 non-smoking patients. What is the statistical power with this sample size? Please comment and discuss in the manuscript.
In the Materials and Methods section, include the appropriate information about reactives, kits, and PCR protocols followed in the nasopharyngeal swabs for COVID-19 diagnosis.
My major concern is that the analyses on both day 1 and day seven have not been adjusted for other covariates. The main covariate of interest is obesity; in this regard, the authors only stated that the mean BMI was 30.15 ± 6.34 kg/m2 (range 17.9-45.0), placing the average patient in the obesity category (BMI ≥ 30kg/m2). Previous investigations have reported that obesity is the most critical comorbidity related to poor or bad outcomes, even more important than tobacco smoking. According to the Body Mass Index (BMI), obesity is classified into grades: Grade I (30.0 to 34.9 kg/m²), Grade II (35.0 to 39.9 kg/m²), and Grade III (40.0 kg/m² or higher). Were there differences according to this variable stratifying by grades? Please analyze or reanalyze if needed.
Another covariate that should be considered in an adjusted analysis is age, since the included patients range from 27 to 86 years. The authors should consider a logistic regression model.
In addition, given that most of the analyses for day 1 were negatives, I suggest moving most of them to the supplementary information file. The current state is unnecessarily repetitive with the text.
Regarding the inflammatory and hematological parameters, many others have reported the relevance of emergent systemic inflammation indices (NHL, NLR, RDW, SII, and SIRI, among others), which have a predictive power of severe COVID-19, IMV support, and low survival probability during hospitalization by COVID-19 in patients instead the individual clinical markers. The authors must make these comparisons and discuss properly. Some examples of those investigations can be consulted on PMIDs: 36341268, 37550520, 36059829.
Authors should include information about other pollutant exposures, mainly biomass-burning smoke, which has been clearly described as an environmental worsening factor in COVID-19 patients.
A paragraph describing the study's main limitations should be included at the end of the discussion.
The discussion section is highly verbose. Please fit the most important information related to the study (design and main results).
Something similar happens with the conclusion section. Please correct it.
Author Response
Thanks for the opportunity to review the manuscript by Djuric and colleagues. In this research, the authors tried to identify early mortality predictors in non-smoking patients with severe COVID-19 by evaluating clinical, laboratory, and oxygenation parameters.
The authors argue an unexpected pattern known as the "smoker's paradox," with lower rates of severe disease among smokers compared to non-smokers, highlighting the need for specific investigation of disease progression in non-smoking populations.
The main idea is interesting, but before I make a final recommendation about its acceptance, I have some concerns for the authors that must be clarified.
Comment 1: The authors included only 59 non-smoking patients. What is the statistical power with this sample size? Please comment and discuss in the manuscript.
Response 1: Thank you for your suggestion. We made corrections and incorporated statistical power with this sample size.
Comment 2: In the Materials and Methods section, include the appropriate information about reactives, kits, and PCR protocols followed in the nasopharyngeal swabs for COVID-19 diagnosis.
Response 2: Thank you for your suggestion. We incorporated requested details.
Comment 3: My major concern is that the analyses on both day 1 and day seven have not been adjusted for other covariates. The main covariate of interest is obesity; in this regard, the authors only stated that the mean BMI was 30.15 ± 6.34 kg/m2 (range 17.9-45.0), placing the average patient in the obesity category (BMI ≥ 30kg/m2). Previous investigations have reported that obesity is the most critical comorbidity related to poor or bad outcomes, even more important than tobacco smoking. According to the Body Mass Index (BMI), obesity is classified into grades: Grade I (30.0 to 34.9 kg/m²), Grade II (35.0 to 39.9 kg/m²), and Grade III (40.0 kg/m² or higher). Were there differences according to this variable stratifying by grades? Please analyze or reanalyze if needed.
Response 3: Thank you for your suggestion. We made and incorporated additional statistical analysis (paragraph 3.4. and Tables 3-5).
Comment 4: Another covariate that should be considered in an adjusted analysis is age, since the included patients range from 27 to 86 years. The authors should consider a logistic regression model.
In addition, given that most of the analyses for day 1 were negatives, I suggest moving most of them to the supplementary information file. The current state is unnecessarily repetitive with the text.
Response 4: Thank you for your suggestion. We made and incorporated additional statistical analysis.
Comment 5: Regarding the inflammatory and hematological parameters, many others have reported the relevance of emergent systemic inflammation indices (NHL, NLR, RDW, SII, and SIRI, among others), which have a predictive power of severe COVID-19, IMV support, and low survival probability during hospitalization by COVID-19 in patients instead the individual clinical markers. The authors must make these comparisons and discuss properly. Some examples of those investigations can be consulted on PMIDs: 36341268, 37550520, 36059829.
Response 5: Thank you for your suggestion. We made corrections.
Comment 6: Authors should include information about other pollutant exposures, mainly biomass-burning smoke, which has been clearly described as an environmental worsening factor in COVID-19 patients.
Response 6: Thank you for your suggestion. We had no information in about other pollutant exposures taking into consideration emergency regarding severe COVID-19 patients however we put your comment into limitations of this study.
Comment 7: A paragraph describing the study's main limitations should be included at the end of the discussion.
Response 7: Thank you for your suggestion. We made corrections.
Comment 8: The discussion section is highly verbose. Please fit the most important information related to the study (design and main results).
Response 8: Thank you for your suggestion. We made corrections.
Comment 9: Something similar happens with the conclusion section. Please correct it.
Response 9: Thank you for your suggestion. We made corrections.
Reviewer 3 Report
Comments and Suggestions for Authors
This study investigates the factors predicting mortality in non-smoking patients suffering from severe COVID-19, a distinct and often overlooked subgroup. The results emphasize the importance of clinical scores (notably the SOFA score), oxygenation levels, and inflammatory markers for prognosis. Nevertheless, enhancing clarity, refining the discussion, and improving language are necessary to boost the manuscript’s overall impact and readability.
- The observation that total cholesterol levels upon admission were notably elevated in non-survivors is intriguing and suggests a potential prognostic significance of lipid metabolism in severe COVID-19 cases. Nonetheless, it is crucial to recognize that the sample size for patients with lipid data was quite limited (just 12 and 7 in the respective groups), which constrains the statistical validity of this finding. This limitation should be more clearly acknowledged in the manuscript.
- The biochemical parameters, specifically lipids, did not exhibit significant differences on day seven. Other studies have reported changes in lipid profiles, particularly cholesterol and triglycerides, during the progression of severe COVID-19.
- The sentence on page 12, lines 304–307, is difficult to follow and should be rewritten for clarity.
- The discussion section is too lengthy and features a lot of redundancies. It would improve with more concise editing to keep readers engaged and emphasize key findings and their implications.
Author Response
Comment 1: This study investigates the factors predicting mortality in non-smoking patients suffering from severe COVID-19, a distinct and often overlooked subgroup. The results emphasize the importance of clinical scores (notably the SOFA score), oxygenation levels, and inflammatory markers for prognosis. Nevertheless, enhancing clarity, refining the discussion, and improving language are necessary to boost the manuscript’s overall impact and readability.
Response 1: Thank you for your suggestion. We made corrections.
Comment 2: The observation that total cholesterol levels upon admission were notably elevated in non-survivors is intriguing and suggests a potential prognostic significance of lipid metabolism in severe COVID-19 cases. Nonetheless, it is crucial to recognize that the sample size for patients with lipid data was quite limited (just 12 and 7 in the respective groups), which constrains the statistical validity of this finding. This limitation should be more clearly acknowledged in the manuscript.
Response 2: Thank you for your suggestion. We put your comment into limitations of this study.
Comment 3: The biochemical parameters, specifically lipids, did not exhibit significant differences on day seven. Other studies have reported changes in lipid profiles, particularly cholesterol and triglycerides, during the progression of severe COVID-19.
Response 3: Thank you for your suggestion. We put your comment into discusion.
Comment 4: The sentence on page 12, lines 304–307, is difficult to follow and should be rewritten for clarity.
Response 4: Thank you for your suggestion. We rewrote the sentence.
Comment 5: The discussion section is too lengthy and features a lot of redundancies. It would improve with more concise editing to keep readers engaged and emphasize key findings and their implications.
Response 5: Thank you for your suggestion. We changed discussion.
Round 2
Reviewer 2 Report
Comments and Suggestions for Authors
I really appreciate that the authors have taken my previous concerns seriously.
Reviewer 3 Report
Comments and Suggestions for Authors
I have carefully reviewed the manuscript titled " Severe COVID-19 in non-smokers: predictive factors and outcomes. " This study highlights the importance of clinical scores, oxygenation levels, and biochemical and inflammatory markers for prognosis. I appreciate the efforts made to address each comment thoroughly. Overall, the revisions have significantly enhanced the quality and clarity of the manuscript. I believe your work now makes a valuable contribution to the literature on COVID-19, particularly concerning non-smoking patients.